# A Comparative Analysis of Microbial Communities in the Rhizosphere Soil and Plant Roots of Healthy and Diseased Yuanyang Nanqi (*Panax vietnamensis*) with Root Rot

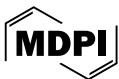

Changyuan Chen [1,2] , Yifan Cheng [1,2], Fangli Zhang [1,2], Saiying Yu [1,2], Xiuming Cui [2,]* and Yuanshuang Wu [1,]*

1   Faculty of Life Science and Technology, Kunming University of Science and Technology, Kunming 650500, China; chenchangyuan618@163.com (C.C.); 20233118004@stu.kust.edu.cn (Y.C.); 13775529603@163.com (F.Z.); saiying_kust@163.com (S.Y.)
2   Yunnan Key Laboratory of Sustainable Utilization of Panax Notoginseng, Kunming 650500, China
*   Correspondence: sanqi37@vip.sina.com (X.C.); wuys@kust.edu.cn (Y.W.)

**Abstract:** Microbial communities are not only an important indicator of soil status but also a determinant of plant nutrition and health levels. Loss of microbial community ecosystem control can directly lead to microbial disease occurrence. During the process of Yuanyang Nanqi wild imitation planting, root rot diseases frequently occur, seriously affecting their yield and quality. Via amplicon sequencing, this study mainly compared the microbial community composition between the rhizosphere soil and roots of healthy and diseased Yuanyang Nanqi with root rot. The α-diversity showed that the microbial community diversity and abundance in the roots of diseased Yuanyang Nanqi were much lower than those of those in healthy specimens, while no significant difference was found in the rhizosphere soil. The β-diversity showed that the bacterial community in the Gejiu region and the fungal community in the Honghe region were significantly different from those in other regions. The species relative abundance map showed that there was no obvious difference in microbial community composition between the rhizosphere soil and roots of healthy and diseased Nanqi, but in diseased specimens with root rot, the proportions of Pseudomonas and Fusarium increased. Based on a functional prediction analysis of FUNGuild, the results showed that the Nanqi roots were mainly pathological saprophytic type and that their rhizosphere soil was mainly saprophytic type. The microorganisms in the roots of Yuanyang Nanqi tubers with root rot were also isolated and identified through the use of the culture method. The possible pathogenic strains were tested via anti-inoculation, and Fusarium oxysporum was identified as one of the main pathogenic fungi of Nanqi root rot, which was consistent with the amplicon sequencing results. These results will help us understand the change trend of microbial communities in healthy and diseased plants and analyze the pathogens involved, the pathogenesis, and the beneficial microorganisms, which would provide a theoretical basis for effective biological control.

**Keywords:** Yuanyang Nanqi; root rot disease; amplicon sequencing; microbial diversity; pathogenic fungi

## 1. Introduction

*Panax vietnamensis Ha & Grushv.* a perennial herb in *Araliaceae* was initially discovered in 1973 on the Yuling Mountain, Kon Tum Province, located in Central Vietnam [1]. In Vietnam, it is known as sâm NgọcLinh or NgọLinh ginseng, later translated as Vietnamese Ginseng. Vietnamese Ginseng primarily thrives in the border areas between China and Vietnam. In China, it has a natural distribution in Jinping Miao, Yao and Dai Autonomous County, Yuanyang County, Lvchun County, and Yunnan Province [2] and is known as Yuanyang Nanqi. As a Vietnamese and Chinese folk medicine, it is commonly used for the post-surgical recovery of women in labor, as well as for the elderly and those who have recovered from serious illnesses, in order to nourish and strengthen their bodies [3]. In folklore, there are also cases of successful use in treating cervical cancer.

Numerous studies have shown that Yuanyang Nanqi, like other plants such as Panax eg, Ginseng, American ginseng, and Panax notoginseng, is rich in saponin components [4–6]. Surprisingly, the total saponin content of Yuanyang Nanqi is much higher than that of other Panax plants, and the content of octreol saponin is especially high, accounting for more than half of the total [7–10]. Additionally, the main active ingredient, ocotillon-type bead ginsenoside-R2 (Majonoside R2, M-R2), makes up more than 5%, roughly half of the total saponin content. Pharmacological studies have shown that M-R2 exhibits pharmacological activities, such as anticancer, anti-aging, anti-inflammatory, and hepatoprotective effects [11–13].

Due to its important role as a medicinal resource, Nanqi has been extensively cultivated in Yuanyang. However, in recent years, issues such as excessively high density in artificially planted Nanqi, continuous soil cropping, and other problems have led to an alarming surge in microbial diseases, among which the incidence rate of root rot disease, the common soil-borne disease of *Panax* plants [14], generally ranges from 10% to 30%, with the incidence of severe cases reaching as high as 80–90%. This has become a critical factor limiting the development of Nanqi planting, resulting in reduced production. Consequently, identifying the dominant microbe responsible for root rot is essential for the effective management of the disease and the sustainable development of Nanqi.

The culture method is the most common technique for studying the pathogenic microorganisms that cause root rot [15]. While this method is valuable for understanding root rot pathogens, limitations arise due to the inability of most plant endophytic bacteria and some fungi to grow on culture media, restricting the evaluation of the diversity and interactions of endophytic microorganism communities [16,17]. High-throughput sequencing technology currently stands as the most advanced detection method, offering notable advantages such as rapid sequencing, high sensitivity, accuracy, and cost-effectiveness. It has played a pivotal role in the field of microbial ecology in recent years [18]. Therefore, microbiomic techniques, involving high-throughput sequencing methods with bacterial 16S rRNA and fungal ITS sequencing, allow for the comprehensive analysis of the microbial communities in their natural state and maximize the exploration of gene resources [19–21].

In this study, in order to investigate their relationship with the occurrence of Nanqi root rot disease, the Illumina MiSeq high-throughput sequencing method [22] was used to analyze the diversity and compositional changes in fungal communities between the roots and rhizosphere soil of 3–4-year-old healthy and root rot Nanqi plants. The culture method was also used to isolate the microorganisms, and the pathogenic microorganisms were verified via a pathogenic inoculation experiment. This work will provide a theoretical foundation for the effective prevention and control of Nanqi root rot disease.

## 2. Materials and Methods

### 2.1. Sampling Location and Sample Collection

The Yuanyang Nanqi samples were obtained from artificial planting sites located in Yuanyang County, Honghe County, and Gejiu City. Twelve healthy and twelve diseased plants with rhizosphere soil were collected in pairs from each sampling point. These were then divided into four groups (Table 1), namely healthy rhizosphere soil (TJ), diseased rhizosphere soil (TB), healthy plant roots (GJ), and diseased plant roots (GB), with three replicates for each group. Plants suffering from root rot disease exhibit significant wilting and yellowing in the aboveground parts and necrosis and decay in the underground parts. Using a fine brush, the soil adhering to the rhizosphere of severely root-rotted plants and adjacent healthy plants was gently collected in a sterilized Petri dish. Every three soil samples from the same site were mixed together; approximately 5 g of each mixed soil sample and plant roots was used for bacterial 16S and fungal ITS amplification analysis, and the remaining parts were quick-frozen with liquid nitrogen and stored in a refrigerator at −80 °C [23] as a retention sample.

**Table 1.** Geographical coordinates of the sampling sites and plant numbers.

| Sample Type | Plant Status | Latitude | Longitude | Number | Group |
|---|---|---|---|---|---|
| Rhizosphere soil | Healthy | 23°18′04″ N | 102°71′94″ E | HHTJ-1 | HHTJ |
| Rhizosphere soil | Healthy | 23°18′04″ N | 102°71′94″ E | HHTJ-2 | HHTJ |
| Rhizosphere soil | Healthy | 23°18′04″ N | 102°71′94″ E | HHTJ-3 | HHTJ |
| Rhizosphere soil | Disease | 23°18′04″ N | 102°71′94″ E | HHTB-1 | HHTB |
| Rhizosphere soil | Disease | 23°18′04″ N | 102°71′94″ E | HHTB-2 | HHTB |
| Rhizosphere soil | Disease | 23°18′04″ N | 102°71′94″ E | HHTB-3 | HHTB |
| Rhizosphere soil | Healthy | 23°12′72″ N | 102°73′25″ E | YYTJ-1 | YYTJ |
| Rhizosphere soil | Healthy | 23°12′72″ N | 102°73′25″ E | YYTJ-2 | YYTJ |
| Rhizosphere soil | Healthy | 23°12′72″ N | 102°73′25″ E | YYTJ-3 | YYTJ |
| Rhizosphere soil | Disease | 23°12′72″ N | 102°73′25″ E | YYTB-1 | YYTB |
| Rhizosphere soil | Disease | 23°12′72″ N | 102°73′25″ E | YYTB-2 | YYTB |
| Rhizosphere soil | Disease | 23°12′72″ N | 102°73′25″ E | YYTB-3 | YYTB |
| Rhizosphere soil | Healthy | 23°41′43″ N | 103°14′58″ E | GJTJ-1 | GJTJ |
| Rhizosphere soil | Healthy | 23°41′43″ N | 103°14′58″ E | GJTJ-2 | GJTJ |
| Rhizosphere soil | Healthy | 23°41′43″ N | 103°14′58″ E | GJTJ-3 | GJTJ |
| Rhizosphere soil | Disease | 23°41′43″ N | 103°14′58″ E | GJTB-1 | GJTB |
| Rhizosphere soil | Disease | 23°41′43″ N | 103°14′58″ E | GJTB-2 | GJTB |
| Rhizosphere soil | Disease | 23°41′43″ N | 103°14′58″ E | GJTB-3 | GJTB |
| Plant Root | Healthy | 23°18′04″ N | 102°71′94″ E | HHGJ-1 | HHGJ |
| Plant Root | Healthy | 23°18′04″ N | 102°71′94″ E | HHGJ-2 | HHGJ |
| Plant Root | Healthy | 23°18′04″ N | 102°71′94″ E | HHGJ-3 | HHGJ |
| Plant Root | Disease | 23°18′04″ N | 102°71′94″ E | HHGB-1 | HHGB |
| Plant Root | Disease | 23°18′04″ N | 102°71′94″ E | HHGB-2 | HHGB |
| Plant Root | Disease | 23°18′04″ N | 102°71′94″ E | HHGB-3 | HHGB |
| Plant Root | Healthy | 23°12′72″ N | 102°73′25″ E | YYGJ-1 | YYGJ |
| Plant Root | Healthy | 23°12′72″ N | 102°73′25″ E | YYGJ-2 | YYGJ |
| Plant Root | Healthy | 23°12′72″ N | 102°73′25″ E | YYGJ-3 | YYGJ |
| Plant Root | Disease | 23°12′72″ N | 102°73′25″ E | YYGB-1 | YYGB |
| Plant Root | Disease | 23°12′72″ N | 102°73′25″ E | YYGB-2 | YYGB |
| Plant Root | Disease | 23°12′72″ N | 102°73′25″ E | YYGB-3 | YYGB |
| Plant Root | Healthy | 23°41′43″ N | 103°14′58″ E | GJGJ-1 | GJGJ |
| Plant Root | Healthy | 23°41′43″ N | 103°14′58″ E | GJGJ-2 | GJGJ |
| Plant Root | Healthy | 23°41′43″ N | 103°14′58″ E | GJGJ-3 | GJGJ |
| Plant Root | Disease | 23°41′43″ N | 103°14′58″ E | GJGB-1 | GJGB |
| Plant Root | Disease | 23°41′43″ N | 103°14′58″ E | GJGB-2 | GJGB |
| Plant Root | Disease | 23°41′43″ N | 103°14′58″ E | GJGB-3 | GJGB |

## 2.2. Genome DNA Extraction

The total genomic DNA was isolated from the samples via the modified CTAB method [24]. A total of 1000 µL of CTAB cracking solution was added to a 2 mL EP tube; then, a 3 g quantity of root or rhizospheric soil sample was added, and the EP tube was placed in a 65 °C water bath with its contents mixed repeatedly. A mixture of phenol/chloroform/isoamyl alcohol (25:24:1) was added and homogenized thoroughly and then centrifugated at 12,000 rpm for 10 min. The supernatant was collected, and the mixture of chloroform/isoamyl alcohol (24:1) was added, mixed upside down, and centrifuged again at 12,000 rpm for 10 min. The supernatant was aspirated into a 1.5 mL centrifuge tube, and isopropanol was added for precipitation at −20 °C. The sediment was collected through centrifugation at 12,000 rpm for 10 min, and this was then washed twice with 1 mL of 75% ethanol. The final sample was dried in a metal bath at 25 °C and dissolved in ddH$_2$O; approximately 1 µL of RNaseA was added to digest the RNA at 37 °C for 30 min; and 1 µL of Proteinase K was added to digest the protein at 65 °C for 30 min. The DNA sample was stored at −80 °C for later use. Then, agarose gel electrophoresis was used to detect DNA purity and concentration [25].

### 2.3. PCR Amplification

The genomic DNA was diluted, and specific primers were used with barcodes based on the selected sequencing region; 16S was amplified using the 799F-1193R region, ITS was amplified using the V5-V7 region in the soil, and the 1F-4R region was amplified in the roots. All of the PCRs were carried out in 30 μL reactions with 15 μL of Phusion® High-Fidelity PCR Master Mix (New England Biolabs, NE, USA); 0.2 μM of forward and reverse primers; and about 10 ng of template DNA. PCR amplification was performed using a Bio-rad T100 gradient PCR instrument (Bio-rad, CA, USA). The amplified PCR product was electrophoresed on a 2% agarose gel, and the sequence with a main band size between 400 and 450 bp was selected. Gel purification was performed using Thermo Scientific's GeneJET Gel Extraction Kit (Thermo Fisher Scientific, Waltham, MA, USA). The DNA concentration was accurately quantified using a Qubit 2.0 fluorometer (Thermo Fisher Scientific Inc., Waltham, MA, USA).

### 2.4. Illumina MiSeq Sequencing

After the DNA samples were quantified, we sent them to Novogene (Beijing, China) Co., Ltd., where they were randomly fragmented using a Covaris Focused-ultrasonicator (Covaris, Inc., Woburn, MA, USA). We then prepared the library according to the Illumina library construction protocol (Illumina Inc., San Diego, CA, USA), including end repair, tailing, sequencing, purification, and PCR amplification steps. Firstly, the library was constructed, and its insertion fragments were detected using an Agilent 2100 biological analyzer (Agilent, Santa Clara, CA, USA). After standardizing the insertion fragment size (500 bp), the Q-PCR method was used to accurately quantify the library's effective concentration to ensure its quality. Secondly, the reservoir was transformed into a flow unit according to the requirements for the effective concentration and target data volume. Finally, the library was sequenced using the NovaSeq 6000 high-throughput sequencing platform (Illumina, Inc., San Diego, CA, USA) and 250 bp paired-end reads were generated [26].

### 2.5. Statistical Analysis

The raw data obtained from sequencing contain a certain proportion of "dirty" data. The original data were spliced and filtered to obtain effective data. Using Uparse software (Uparse v7.0.1001, http://www.drive5.com/uparse/, accessed on 10 October 2022), we clustered all of the effective tags of all samples, and the sequences were clustered into OTUs (operational taxonomic units) with 97% identity by default [27]. The blast method in Qiime software (Version 1.9.1) was used (http://qiime.org/scripts/assign_taxonomy.html, accessed on 12 October 2022) [28], and the Unit (v7.2) database (https://unite.ut.ee/, accessed on 14 October 2022) [29] was used to perform species annotation analysis and determine the community composition of each sample at each taxonomic level—kingdom, phylum, class, order, family, genus, and species—while calculating the observed OTUs, Chao1, Shannon, Simpson, ACE, and other indexes and drawing a dilution curve using R software (version 2.15.3). Qiime software (version 1.9.1) was used to calculate the UniFrac distance and construct the UPGMA sample clustering tree. PCA and PCoA plots were drawn using R software (version 2.15.3). The differences between the β-diversity index groups were analyzed using R software. FUNGuild is the environmental function database for fungi [30]. Based on the existing literature, the ecological functions of fungi were classified and the FUNGuild database was constructed. Based on the species information obtained via amplicon analysis, the ecological functions of existing species in the literature can be queried.

### 2.6. Isolation Microorganisms in the Roots of Diseased Plants

The roots of the diseased plants were sterilized using conventional tissue separation methods [31], flushing the rotten Nanqi roots with sterile water for 30 min, 75% ethanol for 30 s, 5% sodium hypochlorite solution for 2 min, and sterile water 4–5 times sequentially. Under sterile conditions, different parts of Nanqi internal stems were cut into small pieces

and placed on potato dextrose agar (PDA) medium (Beijing Solarbio Science & Technology Co., Ltd., Beijing, China), with 4 pieces on each plate and 3 replicates; these were then cultured at 28 °C. The status of the microbial strains was observed daily, and the growing fungal strains were selected and placed on the PDA culture medium for purification and culturing. The bacteria were purified and cultured on Luria–Bertani broth (LB broth) culture medium (Sangon Biotech (Shanghai) Co., Ltd., Shanghai, China) using the line marking method, and, after being purified, they were stored in test tubes in a refrigerator at 4 °C. The constitution of the PDA culture medium was 200 g of potato, 20 g of glucose, 18 g of agar, and 1000 mL of distilled water, at a natural pH. LB broth culture medium: tryptone, 10 g; yeast powder, 5 g; sodium chloride, 10 g; and agar A, 15 g, at a natural pH.

### 2.7. Molecular Biological Identification of Fungi Strains

The strains were cultured for 7 days, and the mycelium was collected into a centrifuge tube for subsequent sequence analysis. The total DNA was extracted with Solarbio's fungal genome DNA extraction kit (D2330, Beijing Solarbio Science & Technology Co., Ltd., Beijing, China). The sequences of the internal transcribed spacer regions of ribosomal DNA (rDNA ITS), translation elongation factor 1 (TEF1), and partial RNA polymerase II second subunit gene (RPB2) were amplified. PCR amplification was performed by using the fungal universal primers ITS1/ITS4 [32], EF-1/EF-2 [33], EF-3 [34], EF-22T/EF-22U [35], RPB2-5f2/RPB2-7cr [36], and RBP2-7cf/RBP2-11ar [37]. The total PCR volume was 25 μL: DNA template, 1 μL; 2X SanTaq PCR Mix (containing blue dye), 1.5 μL; DNA template, 1 μL each; and distilled water supplemented to 25 μL. PCR procedure: pre-denaturation at 94 °C for 5 min, denaturation at 94 °C for 30 s, 35 cycles and annealing at 56 °C for 30 s, and holding at 72 °C for 10 min, with 30 cycles. After the reaction, the amplification product was verified through 1% agarose gel electrophoresis and then sent to Bioengineering Co., Ltd. (Shanghai, China) for sequencing. The alignment blast was performed using the NCBI database, and the phylogenetic tree was established with Neighbor Joining in Mega9 software.

### 2.8. Pathogenicity Determination and Reisolation of Pathogens

According to Koch's law, the pathogenicity was determined through in vivo inoculation back to the root system of Yuanyang Nanqi. The mycelium of the pathogen was picked with a sterile needle and inoculated into the roots of healthy Yuanyang Nanqi; the roots were then sprayed with $1 \times 10^6$ spores per milliliter for three repetitions. After inoculation, the above-ground portion of the plant was observed daily, and the disease status of the plant was recorded. If the leaves turned yellow and wilted, we observed whether the roots had lesions. After lesions occurred, the pathogenic microorganism was isolated from the inoculated diseased roots again.

### 2.9. Construction of Phylogenetic Trees

After PCR amplification and sequencing, a homology analysis of the pathogenic strains was performed using two accessible DNA sequence databases (the NCBI gene database (https://www.ncbi.nlm.nih.gov/, accessed on 25 December 2023) and FUSARIUM-ID v.3.0 (http://isolate.fusariumdb.org/blast.php, accessed on 26 December 2023 [38]) for BLASTN. The sequences of strains with high homology were used to construct phylogenetic trees on the combined ITS/TEF1/RBP2 sequence via the maximum parsimony method.

## 3. Results

### 3.1. Quality Analysis of Sequencing Results

After 16S rRNA sequencing, a total of 2,748,799 tags were detected in the sample. After quality control and the removal of chimeras and non-target sequences, 2,102,980 effective tags were obtained, with an average of 58,416 tags per sample. Clustering OTUs with 97% homology resulted in a total of 22,064 OTU sequences. After ITS sequencing, a total of 3,021,575 tags were detected in the sample. After quality control and the removal of

chimeras and non-target sequences, 2,266,360 effective tags were obtained, with an average of 62,954 tags per sample. Clustering OTUs with 97% homology resulted in a total of 13,295 OTU sequences. The sparse curves of all of the sample OTUs tended to saturate at 97% sequence similarity (Figure 1). Therefore, the sequencing depth was sufficient to evaluate the bacterial and fungal community diversity in our sample.

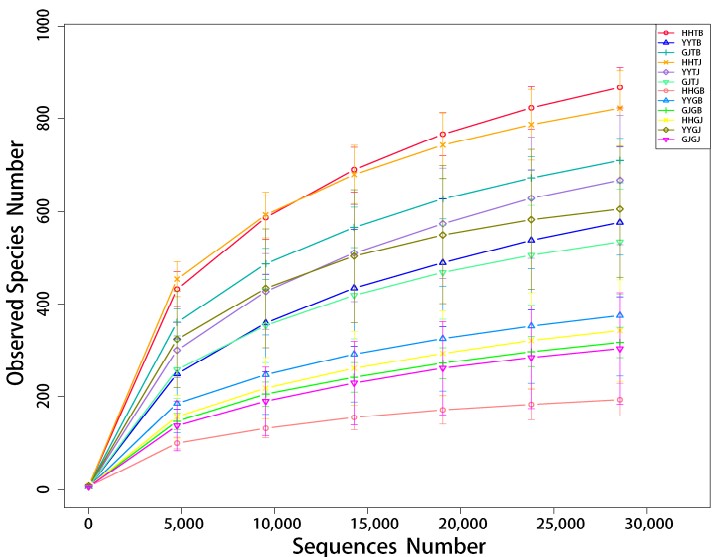

**Figure 1.** Sparse curves for evaluating the abundance of rhizosphere soil and root sequencing data in different regions.

### 3.2. Microbial Diversity of Rhizosphere Soil and Roots in Yuanyang Nanqi OTU Levels

Through OTU analysis (Figure 2), a total of 22,064 bacterial OTU sequences were obtained. Among TB, 1646 OTU sequences were common, with GJTB having the highest number of unique sequences (107); in TJ, 1906 OTU sequences were common, with YYTJ having the highest number of unique sequences (931); in GB, 64 OTU sequences were common, with GJGB having the highest unique number (427); in GJ, 102 OTUs were common, with HHGJ having the highest number of unique occurrences (306). Through OTU analysis, a total of 13,295 fungal OTU sequences were obtained. In TB, 349 OTU sequences were common, with HHTB having the highest number of unique sequences (1106); in TJ, 216 OTU sequences were common, with HHTJ having the highest number of unique sequences (1150); among GB, 167 OTU sequences were common, with YYGB having the highest unique number (514); in GJ, 222 OTUs were common, with YYGJ having the highest number of unique occurrences (989). The results indicate that there were significant differences in the microbial community structures of healthy and diseased plants in the different regions, while, overall, disease reduced the diversity of the bacterial and fungal communities in Nanqi roots.

### 3.3. Bacterial and Fungal α-Diversity

The bacterial and fungal α-diversity index is represented by the ACE index, Chao1 index, and Shannon index (Table 2). For bacterial α-diversity analyses, it was shown that there were significant differences in the bacterial α-diversity in diseased Nanqi roots from different regions of Yuanyang Nanqi. The bacterial ACE index and Chao1 index in the roots of GJGB were significantly higher than those of HHGB and YYGB ($p < 0.05$), with HHGB having the lowest bacterial Shannon index compared to the other regions ($p < 0.05$). The results also showed that the ACE index and Chao1 index of bacteria in the roots of GJGB were significantly higher than those of GJGJ, contrary to the results of HHGB and YYGB.

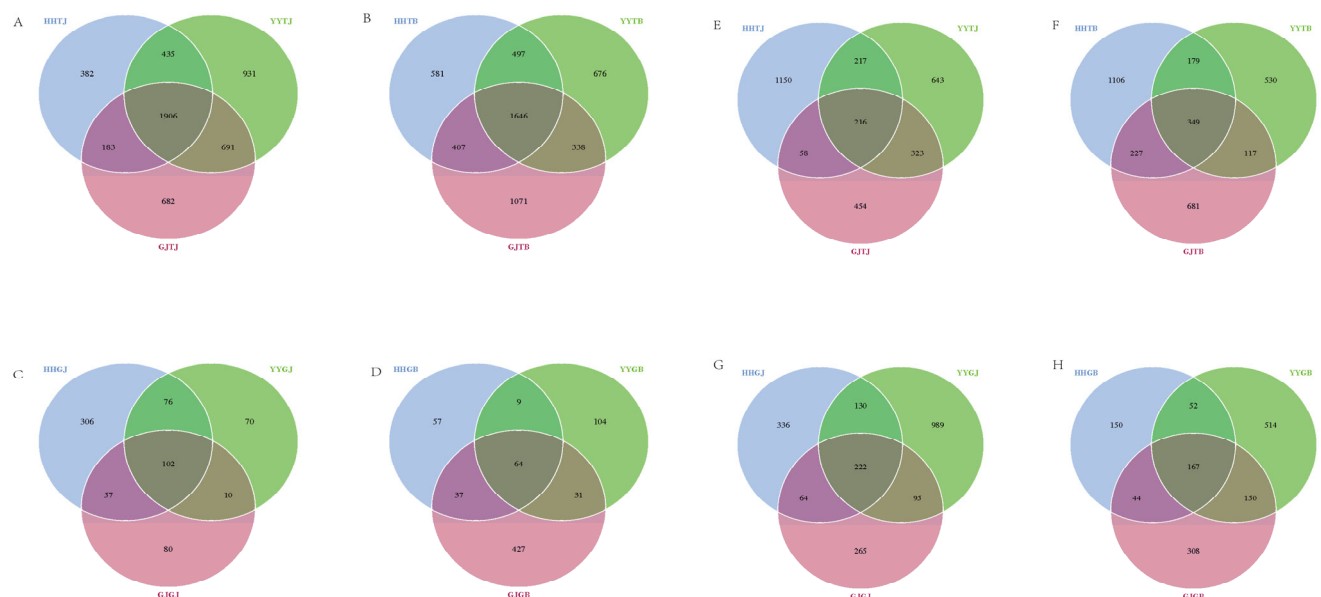

**Figure 2.** Number of 16S bacterial OTUs in the rhizosphere soil of healthy plants in different regions (**A**); number of 16S bacterial OTUs in the roots of healthy plants (**C**); number of ITS fungal OTUs in the rhizosphere soil of healthy plants (**B**); number of ITS fungal OTUs in the roots of healthy plants (**D**); number of 16S bacterial OTUs in the rhizosphere soil of diseased plants in different regions (**E**); number of OTUs of 16S bacteria in the roots of diseased plants (**G**); number of ITS fungal OTUs in the rhizosphere soil of diseased plants (**F**); number of ITS fungal OTUs in the roots of healthy plants (**H**).

**Table 2.** α-diversity of soil and root bacterial and fungal communities in Yuanyang Nanqi.

|  |  | **Bacteria** |  |  |  | **Fungi** |  |  |  |
|---|---|---|---|---|---|---|---|---|---|
|  |  | **OTUs** | **ACE** | **Chao1** | **Shannon** | **OTUs** | **ACE** | **Chao1** | **Shannon** |
| Soil | GJTB | 3462 | 2564.333 | 2521.612 | 9.527 | 1374 | 899.728 | 854.927 | 5.446 |
|  | GJTJ | 3462 | 2206.305 | 2142.524 | 8.145 | 1051 | 694.534 | 659.495 | 4.305 |
|  | HHTB | 3131 | 2756.736 | 2677.472 | 8.566 | 1861 | 1088.916 | 1044.913 | 5.706 |
|  | HHTJ | 2907 | 2215.109 | 2170.085 | 8.387 | 1641 | 963.598 | 944.412 | 5.941 |
|  | YYTB | 3157 | 2492.986 | 2495.198 | 8.988 | 1175 | 821.936 | 768.858 | 3.893 |
|  | YYTJ | 3963 | 2763.247 | 2697.771 | 9.476 | 1399 | 920.027 | 862.295 | 4.353 |
| Root | GJGB | 559 | 420.871 | 409.548 | 1.374 | 669 | 453.300 | 437.215 | 4.137 |
|  | GJGJ | 249 | 207.837 | 185.174 | 0.447 | 646 | 435.469 | 422.407 | 4.009 |
|  | HHGB | 167 | 127.812 | 113.063 | 0.383 | 413 | 240.323 | 236.925 | 3.774 |
|  | HHGJ | 541 | 406.055 | 385.912 | 1.487 | 752 | 491.033 | 505.245 | 4.094 |
|  | YYGB | 208 | 153.551 | 132.562 | 1.110 | 878 | 525.474 | 518.450 | 4.003 |
|  | YYGJ | 258 | 196.795 | 177.253 | 0.452 | 1436 | 691.628 | 675.902 | 5.183 |

For fungal α-diversity analysis, the fungal ACE index and Chao1 index of HHTB in the rhizosphere soil were shown to be higher than those of YYTB and GJTB, with YYTB having the lowest fungal Shannon index. The α-diversity index of YYGB was significantly different from GJGB and HHGB ($p < 0.01$). The fungal ACE index and Chao1 index of HHGB in the roots were lower than those of YYGB and GJGB. The fungal ACE index and Chao1 index of YYGJ were higher than those of HHGJ and GJGJ. The ACE index and Chao1 index of fungi in the roots of HHGB and YYGB were significantly higher than those of HHGJ and YYGJ, respectively, whereas there was no significant difference in the ACE index and Chao1 index between GJGJ and GJGB.

## 3.4. Relative Abundance of Bacterial and Fungal Genera

There were differences in the abundance of each bacterial and fungal genus in the healthy and diseased roots and rhizosphere soils of Yuanyang Nanqi. According to Figure 3A, it can be seen that the dominant bacterial genera of the rhizosphere soil of healthy plants were *unidentified_Cyanobacteria* and *Ralstonia*, which accounted for 33% and 9%, respectively. The dominant bacterial genera in more than 1% of the rhizosphere soil of diseased Yuanyang Nanqi were *Pseudomonas*, *Acidibacter*, *unidentified_Burkholderiaceae*, *Bradyrhizobium*, *unidentified_Gammaproteobacteria*, *unidentified_Rickettsiales*, *Sphingomonas*, and *Pseudogulbenkiania*. The dominant bacterial genera of the samples within the roots of healthy Nanqi in Yuanyang (Figure 3B) were *unidentified_Cyanobacteria* and *Pseudomonas*, which accounted for 98% and 31%, respectively. The dominant bacterial genera in more than 1% of the root samples of diseased Yuanyang Nangqi were *unidentified_Cyanobacteria*, *Pseudomonas*, *Serratia*, *unidentified_Burkholderiaceae*, *Anaerosinus*, *unidentified_Rickettsiales*, *Bradyrhizobium*, *Acidibacter*, *unidentified_Gammaproteobacteria*, *Pseudolabrys*, *Sphingomonas*, *Xylophilus*, *Haliangium*, *Pseudogulbenkiania*, and *Ralstonia*. *Pseudomonas* strains accounted for a large proportion in the diseased plants.

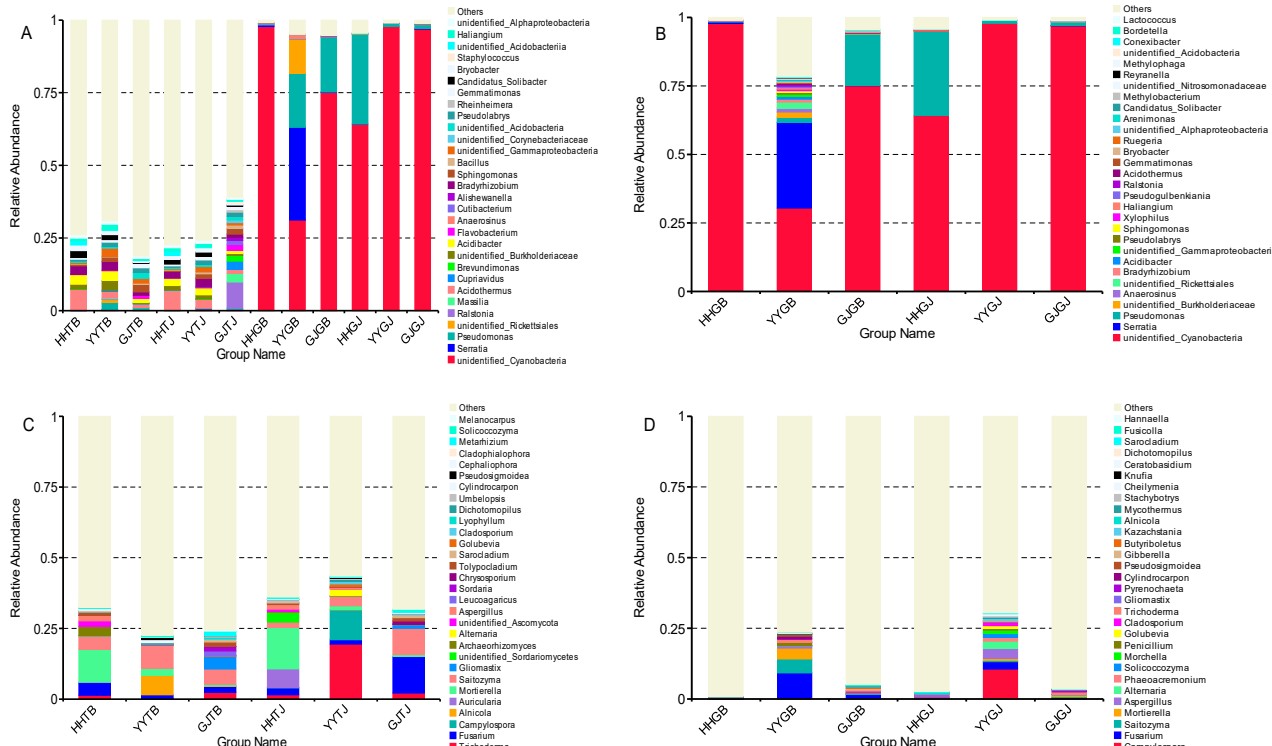

**Figure 3.** 16S sequencing of bacteria in the top 30 genera (**A**,**B**) and ITS sequencing of fungi in the top 30 genera (**C**,**D**) in the roots and rhizosphere soil of Yuanyang Nanqi.

Based on Figure 3C, it can be seen that the dominant fungal genera in the inter-root rhizosphere soil of healthy Yuanyang Nanqi were *Trichoderma*, *Fusarium*, *Campylospora* and *Mortierella*, accounting for 19%, 13%, 11%, and 15%, respectively. The dominant fungal genera in the rhizosphere soil of diseased Yuanyang Nanqi with more than 1% disease were *Fusarium*, *Alnicola*, *Mortierella*, *Saitozyma*, *Gliomast*, *Archaeorhizomycesix*, *Leucoagaricus*, *Sordaria*, *Tolypocladium*, *Cylindrocarpon*, *Pseudosigmoidea*, and *Cladophialophora*. The dominant fungal genera within the roots of healthy Yuanyang Nanqi (Figure 3D) were *Campylospora*, *Aspergillus*, and *Alternaria*, with percentages of 11%, 4%, and 2%. The dominant fungal genera in the roots of Yuanyang Nanqi with root rot were *Fusarium*, *Saitozyma*, *Mortierella*, *Cylindrocarpon*, *Pseudosigmoidea*, *Gibberella*, and *Butyriboletus*. The proportion of *Fusarium* strains in the diseased plants became larger.

### 3.5. β-Diversity Analysis

This study used a UPGMA clustering tree and PCA to cluster Yuanyang Nanqi based on unweighted Unifrac UPGMA (Figure 4). Clustering analysis was carried out using OTU and an unweighted Unifrac Distance matrix, and the results were combined with the relative abundance histogram at the gate taxon level. The phylogenetic tree shows that the bacterial community is divided into three major groups (Figure 4A,B) and the fungal community is divided into two major groups (Figure 4C,D), each containing root and rhizosphere soil samples from different regions. A PCA of the bacteria and fungi showed that, for bacterial communities (Figure 4E), the first and second principal components explained the variances of 21.74% and 11.84%, respectively. For fungal communities (Figure 4F), the first and second principal components explained the variances of 9.86% and 7.95%, respectively. The results showed that the bacterial community in the roots had significant aggregation compared to that in the rhizosphere soil, and the fungal community in the roots also had significant aggregation compared to the fungal community in the rhizosphere soil. The bacterial communities of GJTB and GJTJ bore little resemblance to those of HHTB, HHTJ, YYTB, and YYTJ. The fungal communities of HHTB and HHTJ demonstrated little similarity with those of HHTB, HHTJ, YYTB, and YYTJ.

### 3.6. Functional Prediction of Fungal Communities

Using FUNGuild, a functional prediction analysis of plant roots and rhizospheric soil fungi in Yuanyang Nanqi was performed [39]. Species with confidence levels of "Highly Probable" and "Probable" were screened and visualized to obtain fungal ecological nutrient types and their proportions in different samples. The TrophicMode results show that the fungal communities in the roots and non-rhizosphere soils of Yuanyang Nanqi mainly predict nine nutrient types (Figure 5A): *Pathotroph-Saprotroph*, *Saprotroph*, *Symbiotroph*, *Pathotroph-Saprotroph-Symbiotroph*, *Pathotroph*, *Pathotroph-Symbiotroph*, *Saprotroph-Symbiotroph*, *Pathogen-saprotroph-symbiotroph*, and *Saprotroph-Pathotroph-Symbiotroph*. The nutrient type with the highest proportion of roots is the pathological saprophytic nutrient type. The highest proportion of rhizospheric soil is the saprophytic nutrient type. Based on Guild's results, a total of 24 ecologions were predicted for both plant roots and rhizosphere soils, and their relative abundances are shown in Figure 5B. Among these, the dominant co-site groups in rhizosphere soil were *Plant_Pathogen-Soil-Wood_Saprotroph*, *Saprotroph*, *Ectomycor-rhizal*, *Fungal_Parasite*, *Plant_Pathogen-Soil-Wood_Saprotroph*, and *Saprotroph*.

### 3.7. Molecular Identification Results of Microorganisms Separated via the Culture Method

The isolated strains were sequenced, and the sequences were compared using the NCBI gene database; the reference strains were compared, as shown in Table 3. The analysis showed that the strains of *Fusarium*, *Discospora*, *Echinococcus*, *Pseudomonas*, and *Epiphyllum* may cause diseases in plants.

### 3.8. Anti-Inoculation Test Results and Phylogenetic Tree of Pathogenic Fungi

The HHGBZHEN1 (*Diaporthe*) [40], HHGBZHEN7 (*Fusarium*) [41], and HHGBZHEN4 (*Ilyonectria*) [42] strains were inoculated in the root system of healthy 2–3-year-old Yuanyang Nanqi. The isolation and purification of roots from diseased plants after inoculation led to obtaining bacteria identical to the inoculated strain, proving that HHGBZHEN7 is the pathogen of the disease (Figure 6).

The BLASTN analysis of the NCBI database indicated that the ITS, TEF1, and RBP2 sequences have the highest nucleotide homology in *F. oxysporum* ITS (MG564294.1), TEF1 (MN507110.1), and RBP2 (MT188716.1). The BLASTN analysis of the FUSARIUM-ID v.3.0 database indicated that the ITS, TEF1, and RBP2 sequences have the highest nucleotide homology in *F. oxysporum* ITS (NR182848.1), TEF1 (NRRL26875), and RBP2 (JX171625.1). The gene sequences were submitted to GenBank (Accession numbers OR726039.1 (ITS)). Phylogenetic trees based on the combined ITS/TEF1/RBP2 sequence were constructed via the maximum parsimony method, as shown in Figure 7. These results demonstrate

that HHGBZHEN7 is *Fusarium oxysporum. Fusarium oxysporum* is one of the pathogens that causes root rot disease in Yuanyang Nanqi.

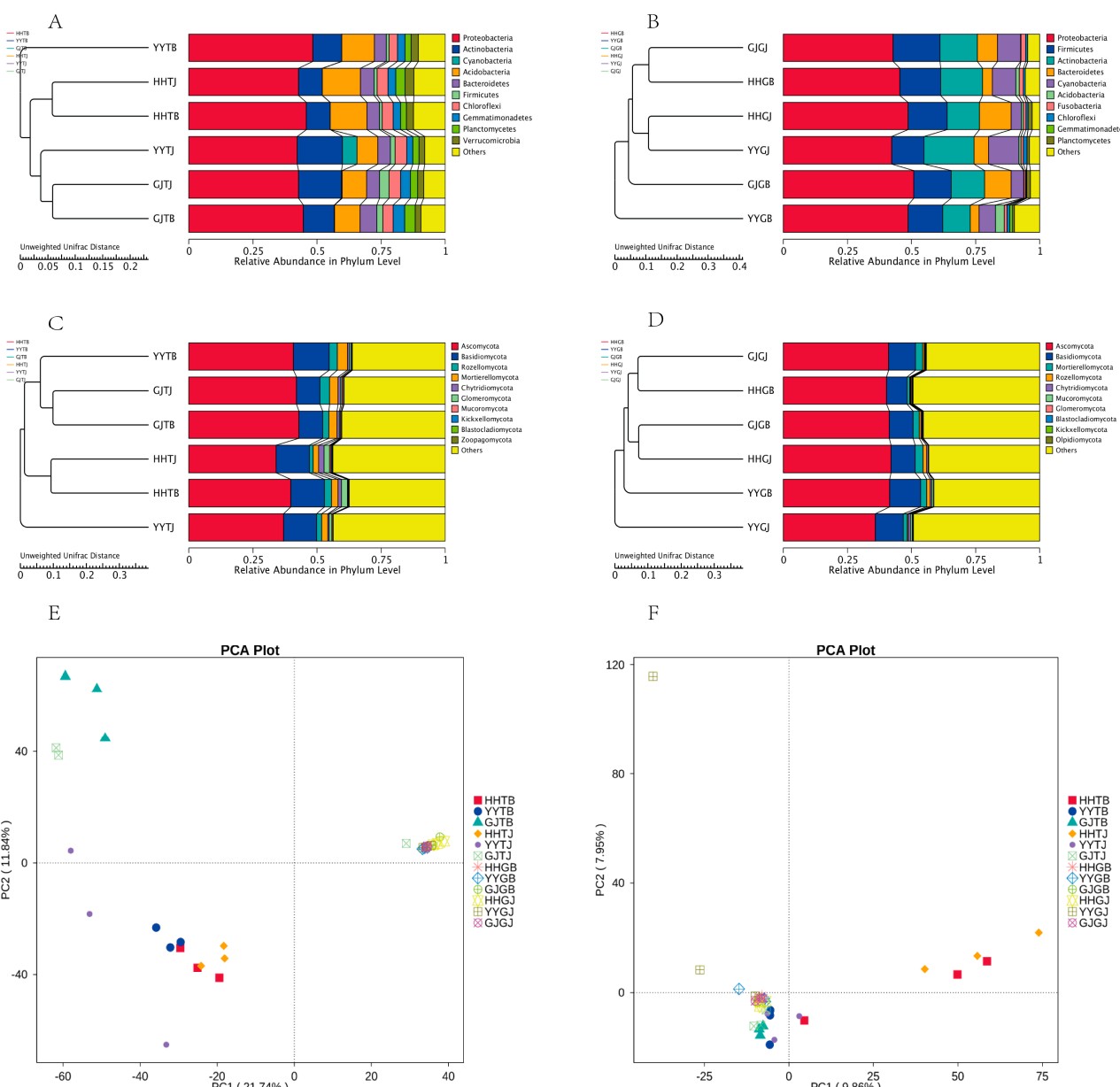

**Figure 4.** Analysis of the microbial community of Nanqi in Yuanyang using unweighted Unifrac. UPGMA cluster analysis of bacteria (**A**,**B**) and fungi (**C**,**D**) in soil and roots; the left side is the UPGMA cluster tree structure, and the right side is the distribution of the species relative abundance of each sample at the phylum level. PCA of rhizosphere soil and roots' bacteria (**E**) and fungi (**F**) in Nanqi.

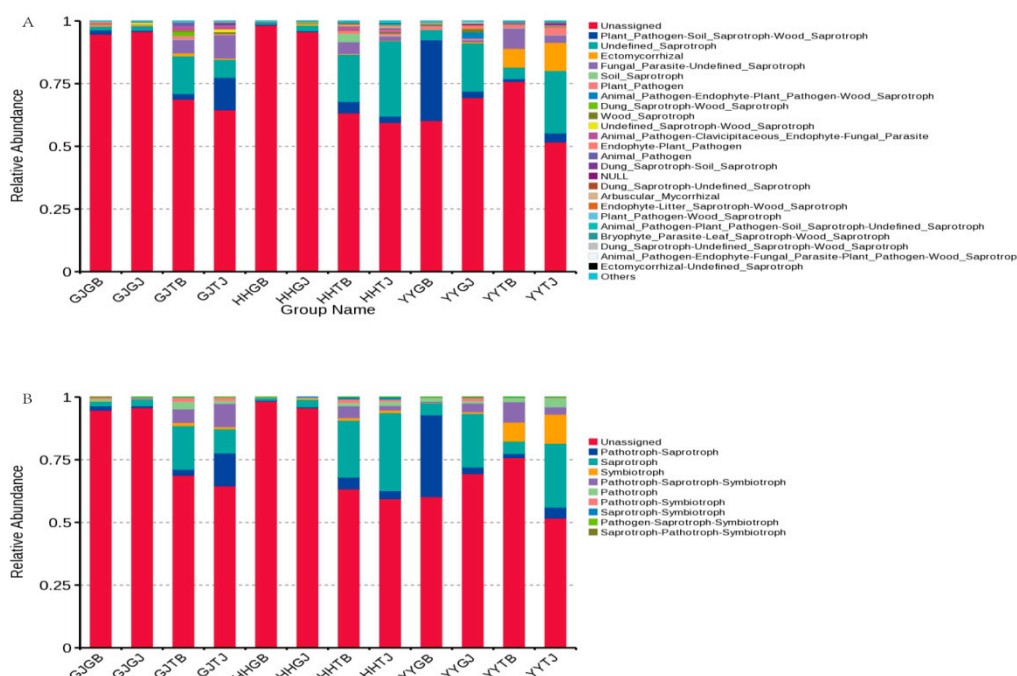

**Figure 5.** TrophicMode (**A**) and Guild results (**B**) for the functional prediction of root and rhizosphere soil fungal communities.

**Table 3.** Isolation of endophytic fungi and bacteria from Yuanyang Nanqi via the culture method.

| Number | Type Strain | Reference Strain | Homology |
|---|---|---|---|
| GJGB ZHEN2 | MT561402.1 | *Chaetomium cochliodes* | 99.81% |
| GJGB ZHEN3 | MT561402.1 | *Chaetomium cochliodes* | 99.45% |
| GJGB ZHEN5 | MT561402.1 | *Chaetomium cochliodes* | 100% |
| GJGB ZHEN11 | MN689717.1 | *Fusarium solani* | 97.01% |
| GJGB ZHEN12 | MN689717.1 | *Fusarium solani* | 94.82% |
| GJGB ZHEN13 | MH856055.1 | *Fusarium tonkinense* | 96.94% |
| GJGB ZHEN14 | MN689717.1 | *Fusarium solani* | 99.63% |
| HHGB ZHEN1 | MN816432.1 | *Diaporthe fusicola* | 99.82% |
| HHGB ZHEN2 | MN816432.1 | *Diaporthe fusicola* | 98.40% |
| HHGB ZHEN3 | OP090371.1 | *Fusarium oxysporum* | 99.23% |
| HHGB ZHEN4 | NR_152889.1 | *Ilyonectria leucospermi* | 99.42% |
| HHGB ZHEN6 | KR364584.1 | *Fusarium oxysporum* | 97.90% |
| HHGB ZHEN7 | MG564294.1 | *Fusarium oxysporum* | 99.79% |
| HHGB ZHEN9 | MT606194.1 | *Colletotrichum* sp. | 97.44% |
| HHGB ZHEN11 | MT476912.1 | *Trametes hirsuta* | 99.33% |
| HHGB ZHEN15 | MG543786.1 | *Galactomyces pseudocandidus* | 99.64% |
| HHGB ZHEN16 | MW221103.1 | *Galactomyces pseudocandidus* | 98.30% |
| YYGB ZHEN2 | OP237439.1 | *Fusarium solani* | 94.40% |
| YYGB ZHEN3 | MH397492.1 | *Fusarium* sp. | 99.81% |
| YYGB ZHEN4 | MF467275.1 | *Fusarium oxysporum* | 99.81% |
| YYGB ZHEN5 | MT251175.1 | *Fusarium falciforme* | 99.63% |
| YYGB ZHEN6 | MT605584.1 | *Fusarium solani* | 100% |
| YYGB ZHEN7 | MT605584.1 | *Fusarium solani* | 97.25% |
| YYGB ZHEN8 | MG650069.1 | *Fusarium* sp. | 96.58% |
| YYGB ZHEN9 | MT032635.1 | *Fusarium oxysporum* | 99.81% |
| YYGB ZHEN10 | MK372218.1 | *Penicillium* sp. | 99.81% |

**Table 3.** *Cont.*

| Number | Type Strain | Reference Strain | Homology |
|---|---|---|---|
| GJGJ ZHEN3 | MN341788.1 | *Nemania diffusa* | 99.45% |
| GJGJ ZHEN4 | MK372218.1 | *Penicillium* sp. | 99.81% |
| HHGJ ZHEN1 | OK030894.1 | *Colletotrichum* sp. | 100% |
| HHGJ ZHEN2 | MT322235.1 | *Colletotrichum gigasporum* | 100% |
| HHGJ ZHEN3 | MT588864.1 | *Chaetomium globosum* | 99.63% |
| HHGJ ZHEN4 | OK030894.1 | *Colletotrichum* sp. | 99.24% |
| GJGB XI2 | KY689942.1 | *Pseudomonas* sp. | 99.71% |
| GJGB XI3 | MT568560.1 | *Raoultella ornithinolytica* | 99.93% |
| GJGB XI4 | MK834812.1 | *Pseudomonas* sp. | 99.61% |
| GJGB XI5 | MT568560.1 | *Raoultella ornithinolytica* | 99.93% |
| HHGB XI1 | MT341797.1 | *Pseudomonas* sp. | 100% |
| HHGB XI4 | CP027561.1 | *Pseudomonas fluorescens* | 99.78% |
| HHGB XI6 | MN826151.1 | *Burkholderia contaminans* | 99.62% |
| HHGB XI7 | MG593863.1 | *Burkholderia* sp. | 99.70% |
| HHGB XI8 | MG571655.1 | *Pseudomonas batumici* | 99.77% |
| HHGB XI9 | MG571655.1 | *Pseudomonas batumici* | 99.70% |
| HHGB XI10 | MT895634.1 | *Paraburkholderia* sp. | 99.55% |
| HHGB XI11 | MT341797.1 | *Pseudomonas* sp. | 99.85% |
| YYGB XI1 | KU950364.1 | *Serratia liquefaciens* | 99.86% |
| GJGJ XI1 | KT695823.1 | *Pseudomonas fluorescens* | 99.85% |
| GJGJ XI2 | KC790251.1 | *Pseudomonas reinekei* | 99.93% |
| GJGJ XI3 | CP024646.1 | *Pseudomonas syringae* | 99.85% |
| HHGJ XI1 | MW959056.1 | *Pseudomonas* sp. | 99.34% |
| HHGJ XI3 | GQ306158.1 | *Burkholderia* sp. | 99.12% |
| HHGJ XI4 | KJ781942.1 | *Rahnella aquatilis* | 99.03% |
| HHGJ XI5 | EU275360.1 | *Rahnella* sp. | 99.77% |
| HHGJ XI7 | MF948895.1 | *Paraburkholderia ginsengiterrae* | 99.46% |

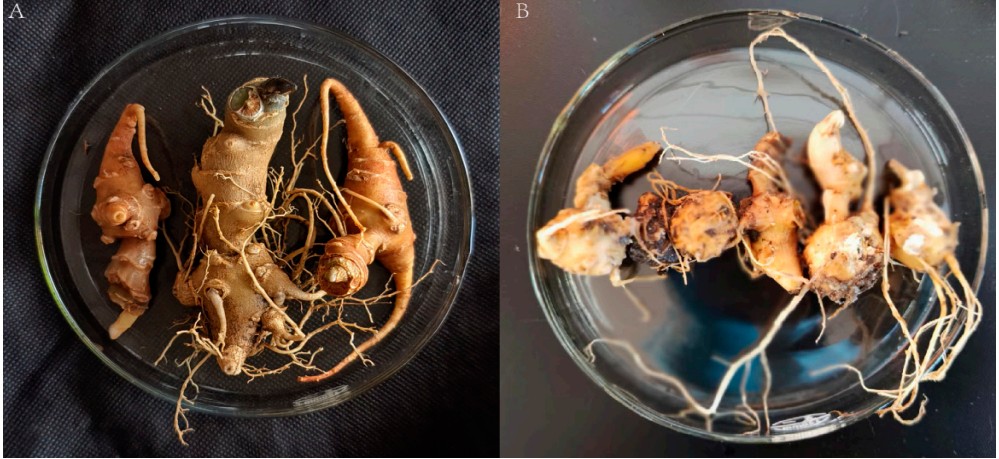

**Figure 6.** Root infection after inoculation with pathogens. Healthy root morphology (**A**); root morphology after infection (**B**).

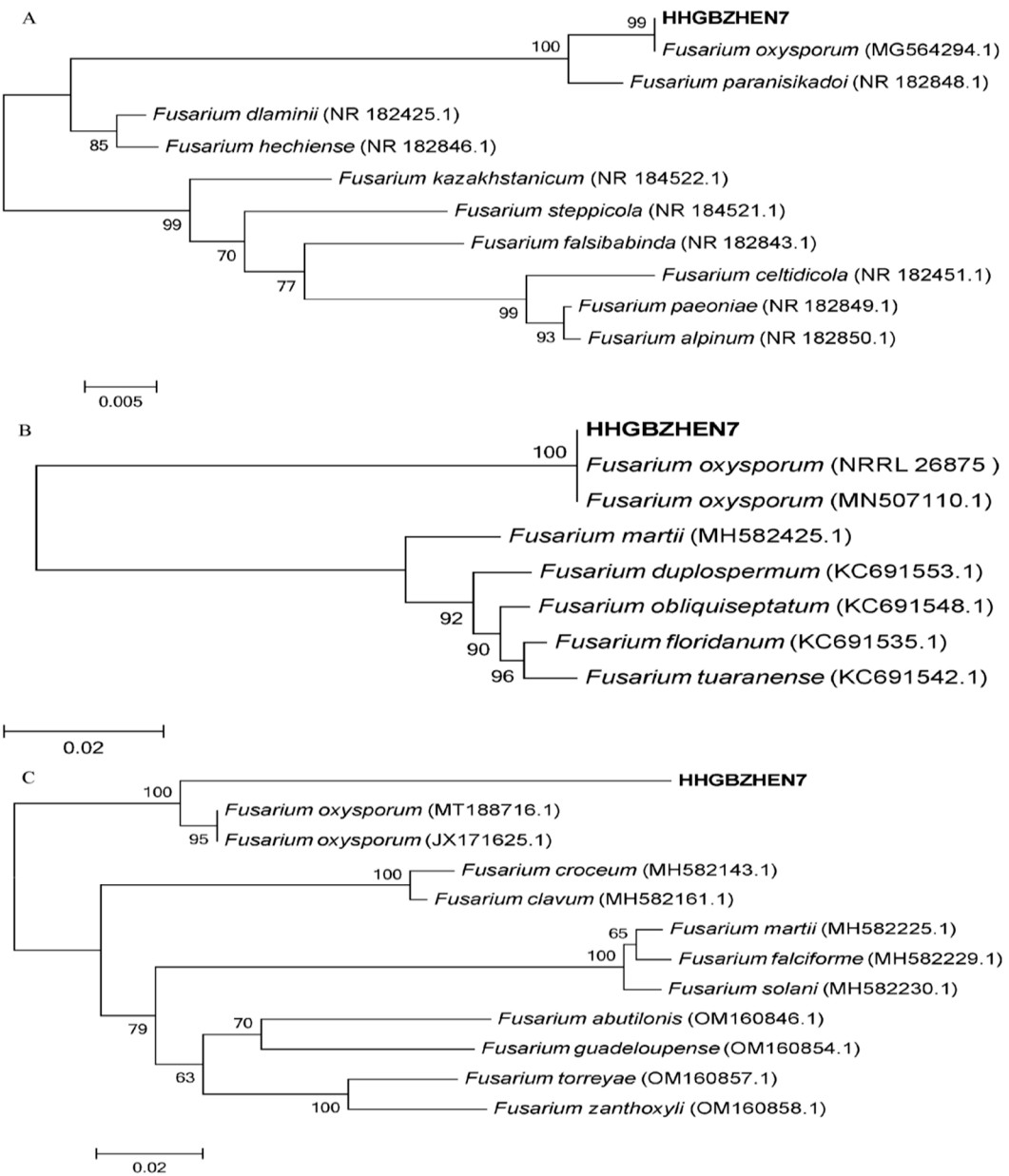

**Figure 7.** The phylogenetic relationship between the endophytic isolate HHGBZHEN7 from Yuanyang Nanqi and other species in GenBank. The adjacency method (NJ) tree was constructed using partial sequence combinations in the ITS (**A**); TEF1 (**B**); and RBP2 (**C**) regions. A bootstrap value with 1000 repeated tests is provided at each node. The GenBank storage number is indicated after the species name.

## 4. Discussion

There are abundant microbial communities of bacteria and fungi in ginseng roots and rhizosphere soils [23]. Complex microbial interactions in the roots have important effects on the health and growth of plants [43]. However, most previous studies on plant diseases have only analyzed the effect of strains on the disease using the culture method, which has certain limitations. The Illumina MiSeq high-throughput sequencing method used in this study overcomes this limitation and can obtain thousands of sequences at the same time, identifying many more microorganisms [44]. With this method, we have the chance to systematically compare the microbial communities in the rhizosphere soil and roots between diseased and healthy Nanqi, and to better understand the changes in microorganisms during root rot.

OTU analysis showed that there are significant differences in the microbial community structure of healthy and diseased plants in different regions, but, overall, the disease reduced the diversity of bacterial and fungal communities in Nanqi roots. This result is consistent with previous research, which found that geographical factors could lead to certain differences in the composition and abundance of soil microbial communities in different geographical locations [45]. There are also studies indicating that the specific changes in soil microbial community diversity and structure are related to differences in soil structure and plant type [46].

In α-diversity analysis, it was also found that the ACE and Chao1 indexes of bacteria in the roots of GJGB were significantly higher than those of GJGJ, contrary to the results of HHGB and YYGB and that the ACE and Chao1 indexes of fungi in the roots of HHGB and YYGB were significantly higher than those of HHGJ and YYGJ, respectively, with no significant difference in the ACE and Chao1 indexes of GJGJ and GJGB. This may be due to the influence of soil conditions, microbial structures, or different main pathogens. For example, research has found that the microbial diversity of the rhizosphere soil of banana with wilt disease is higher than that of healthy soil, which is probably due to the sustained impact of the disease leading to changes in soil physicochemical properties and microbial community structure [47]. Another study also found that the invasion of pathogenic microorganisms can disrupt the original microbial ecological balance in the rhizosphere soil, leading to an abnormal increase or decrease in the number of some microorganisms. Before a new balance is formed, the microbial diversity will temporarily increase [48]. In addition, though research on root rot found that the pathogenic fungus is fusarium, Pseudomonas can also cause root soft rot with similar symptoms, which is called bacterial root rot [49]. The pathogenic microorganisms in GJGB may be different from HHGB and YYGB, or they may have different pathological processes.

According to UPGMA cluster and PCA β-diversity analyses (Figures 4 and 5), there are some differences in the dominant microbial populations in different geographic locations, which is consistent with the results showing that geographic factors dominate in the composition of soil microbial communities. Meanwhile, in the root community, the bacterial genera *unidentified_Cyanobacteria*, *Pseudomonas*, *Serratia*, etc., constitute the largest proportion of the root communities, of which *Pseudomonas* is the dominant bacterial genus. Among the fungal genera, *Fusarium*, *Saitozyma*, *Mortierella*, *Cylindrocarpon*, *Pseudosigmoidea*, and *Gibberella* make up the largest proportion; *Fusarium* is the dominant fungal genus in the rhizosphere community, which is basically consistent with studies of the root communities of other ginseng plants [49–52]. This shows that the strains of these two genera have a greater impact on ginseng root rot. By using FUNGuild to predict the function of fungi in roots and rhizosphere soil, we found that the main predicted nutrient type of fungal groups in Yuanyang Nanqi was pathotrophic saprotroph (Figure 6), in which the dominant common group was *Saprotrophic* bacteria, which increased in diseased plants. According to Wang R. and others in their study of Sophora flavescens, when its symbiotic vegetative fungi are dominant, the host's anti-interference ability is enhanced, while the proportions of pathological vegetative and saprophytic fungi are higher than those of symbiotic vegetative fungi, which very easily cause plant diseases [53]. Meanwhile, the root microbial community of plants affects the soil bacterial community, soil fertility, and crop yield. In addition to the microbiome, other potential factors, including environmental variables, soil environmental factors, nematodes, etc., may also lead to ginseng plant diseases and interactions, which warrants further investigation in the future [54].

The amplicon analysis showed that the main fungal genus among the diseased roots was *Fusarium* and the main bacterial genus was *Pseudomonas*. The same result was also reported for panax root rot [55]. *Pseudomonas* exists in rhizosphere soil and roots, and its content is higher in diseased soil. However, in this study, after preliminary screening, it was found that the isolated *Pseudomonas* was a beneficial bacterium. As in previous studies, *Pseudomonas* strains may be harmful [49] or beneficial bacteria [56,57]. In the future, we should pay attention to this genus. In this study, six possible pathogenic bacteria, *D.*

*fusicola*, *F. oxysporum*, *F. solani*, *Colletotrichum* sp., *I. leucospermi*, and *E. sorghinum*, were isolated. However, based on the analysis of the data from back-inoculation experiments with individual strains of *D. fusicola*, *F. oxysporum*, and *F. solani*, only *F. oxysporum* showed pathogenicity. This may be related to the local humidity and strain interactions and spore concentrations. In addition, in recent studies, the genus *Ilyonectria* was also found to cause ginseng root rot [58], but because there were few strains isolated from this genus in this experiment, we will consider the impact of compound pathogenic bacterial infection on plants and carry out anti-inoculation experiments for several other possible pathogenic bacteria while ensuring humidity.

Existing studies show that most of the pathogenic bacteria causing Chinese herbal ginseng root rot are *Fusarium* sp. [59]. This is consistent with our finding that the *Fusarium oxysporum* strain (HHGBZHEN7) isolated and verified by us is the pathogen causing Nanqi root rot. *Fusarium oxysporum* is a well-known soil-derived plant pathogen [60]. In the process of infection, *Fusarium oxysporum* not only secretes various virulence factors, such as cell wall-degrading enzymes (CWDEs), effectors, and mycotoxins [61,62], which may play an important role in fungal pathogenicity, but must also respond to the external abiotic stresses of the environment and the host. Its mechanism of action on Nanqi remains to be further explored. As far as we know, our study is the first to report on the relationship between the composition and function of rhizosphere soil microorganisms and pathogenic bacteria in Yuanyang Nanqi root rot disease, providing some theoretical basis for the follow-up study of this disease.

## 5. Conclusions

The results of this study showed that (i) the abundance and diversity of microbial communities in the rhizosphere soil and roots of diseased Nanqi were significantly different from those of healthy Nanqi; (ii) *Pseudomonas* was the dominant bacterium and *Fusarium* was the dominant fungus in Nanqi with root rot; and (iii) *F. oxysporum* was one of the pathogenic fungi of Nanqi suffering from root rot disease. In summary, this paper introduced the relationship between the microbial composition of root and rhizosphere soil and the occurrence of diseases in Yuanyang Nanqi, which will help us understand the pathogenesis of microbial diseases and provide a theoretical basis for effective biological control.

**Supplementary Materials:** The following supporting information can be downloaded at: https://www.mdpi.com/article/10.3390/agriculture14050719/s1. Figure S1. Bacteria and fungi *α*-diversity Significance analysis; Table S1. Primers used for PCR and DNA sequencing.

**Author Contributions:** X.C. (Xiuming Cui) and Y.W. designed the research; C.C. (Changyuan Chen) performed most of the research and data analysis; Y.C. performed part of the preparation of the plant material; F.Z. and S.Y. performed part of the separation of microorganisms; C.C. (Changyuan Chen) wrote the original draft; Y.W. and X.C. (Xiuming Cui) contributed to the review and editing of the manuscript. All authors have read and agreed to the published version of the manuscript.

**Funding:** This article was supported by the Capacity Building Project on Sustainable Utilisation of Valuable Traditional Chinese Medicine Resources under the responsibility of the China Academy of Traditional Chinese Medicine (CATCM) (No. 2060302).

**Data Availability Statement:** Data are contained within the article or Supplementary Materials.

**Conflicts of Interest:** All authors declare no conflicts of interest. All authors have read and agreed to the published version of the manuscript.

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
