# Peer review of "A Comparative Analysis of Microbial Communities in the Rhizosphere Soil and Plant Roots of Healthy and Diseased Yuanyang Nanqi (Panax vietnamensis) with Root Rot"

_agriculture, doi:10.3390/agriculture14050719_

Round 1

Reviewer 1 Report

Comments and Suggestions for Authors

Dear editor and corresponding author,

Thank you for the invitation to referee this paper.

The title of your paper promised a very interesting scientific study. You have good data but your results that deserve to be presented in a better way.

Some comments on the manuscript:

English corrections: I converted the manuscript form PDF to WORD and I made some corrections to the English  in the introduction. I used tracked changes so the corresponding author can see the changes. There were more than 10 corrections which is too many. I did not correct the whole document – but it needs a thorough and careful revision (see attached) . Other than English language corrections there were many typing errors:

Lines 198, 200 -  you must be more careful. OTU is correct but OUT appears frequently - this Is it a typing/editing error which should reach me as a reviewer?

16s should be 16S rRNA  or 16S rDNA    -- The “S” is for the Svedberg unit and its symbol is a capital “S” everywhere where you wrote 16s change to 16S .

Results Table 2,

How many sequences from each sample site were analysed ?  Did you normalize the data ? If not normalized how can you make meaningful comparisons? Please revise this in materials and methods and also in the results section.   In Table 2 – what is "Soli" ? The paper needs to be proof read before submitting.

In Fig 2  - In some samples the dominant group is “others”… Analysis at the Phylum level is of little practical use – even more so when “others” dominate. Given that a 16S or ITS approach was used then the correct level of analysis should have been Genus level.  I suggest you adjust all results and analyses accordingly to genus level.

I suggest you remove the current Fig 2 and include a new fig 2 and show the distribution of genera and not phyla.

Materials and Methods is incomplete - the ecological and trophic methodologies and statistical approaches are not described. The results shown in Figs3,  4 and  5 (all very interesting are not in materials and methods) and and barely mentioned in the introduction. Consequently they are not properly discussed.

You have a very nice data set but you have not introduced the subject properly , nor described your methods fully as such your results cannot be verified. My recommendation is a re-write of the paper and a new submission when it is ready.

 Best wishes

Comments on the Quality of English Language

Dear editor and corresponding author,

Thank you for the invitation to referee this paper.

The title of your paper promised a very interesting scientific study. You have good data but your results that deserve to be presented in a better way.

Some comments on the manuscript:

English corrections: I converted the manuscript form PDF to WORD and I made some corrections to the English  in the introduction. I used tracked changes so the corresponding author can see the changes. There were more than 10 corrections which is too many. I did not correct the whole document – but it needs a thorough and careful revision (see attached) . Other than English language corrections there were many typing errors:

Lines 198, 200 -  you must be more careful. OTU is correct but OUT appears frequently - this Is it a typing/editing error which should reach me as a reviewer?

16s should be 16S rRNA  or 16S rDNA    -- The “S” is for the Svedberg unit and its symbol is a capital “S” everywhere where you wrote 16s change to 16S .

Results Table 2,

How many sequences from each sample site were analysed ?  Did you normalize the data ? If not normalized how can you make meaningful comparisons? Please revise this in materials and methods and also in the results section.   In Table 2 – what is "Soli" ? The paper needs to be proof read before submitting.

In Fig 2  - In some samples the dominant group is “others”… Analysis at the Phylum level is of little practical use – even more so when “others” dominate. Given that a 16S or ITS approach was used then the correct level of analysis should have been Genus level.  I suggest you adjust all results and analyses accordingly to genus level.

I suggest you remove the current Fig 2 and include a new fig 2 and show the distribution of genera and not phyla.

Materials and Methods is incomplete - the ecological and trophic methodologies and statistical approaches are not described. The results shown in Figs3,  4 and  5 (all very interesting are not in materials and methods) and and barely mentioned in the introduction. Consequently they are not properly discussed.

You have a very nice data set but you have not introduced the subject properly , nor described your methods fully as such your results cannot be verified. My recommendation is a re-write of the paper and a new submission when it is ready.

 Best wishes.

Reviewer 2 Report

Comments and Suggestions for Authors

Is a very interesting study - however, as the author are study a so important subject - the reader could benefit about some abiotic data especially that the samples were collected from different sites. The reference citation need a strong examination ! I did not found  a clear objective about the goal of study and no hypothesis. Before the line 30 I think that supposed to be "Introduction'.  A strong English editing is required  for example are not clear (line247). the subject is interesting and the data obtained could have a great input on the subject. 

specific comments such as:
- Methodology is not always of great importance if we don’t have the causes for such pathogens. Methodology is a tool that helps to understand – what is the hypothesis ? objective ? What are the main causes for the appearance of such pathogenic microorganisms? What about the abiotic variables that such pathogens appear ? – Theoretical foundation is not of great importance – therefore, it is of great importance CHANGE the title.

- a control from open field sample should be taken – in order to determine the pool of pathogens
- abiotic variable that could be one of the causes for dead or long term survival of pathogens - pool

- the amount of data collected with all the statistical analysis e.g. heat maps not always necessary – too much analysis do not simplify the answer to (a question that had not been asked) goal of the study.
- the manuscript data analysis as well the discussion and the conclusions are missing the focus of the study
The English – should be improved

 the tables and figures:
-Too many figures and statistical analysis in response to a straight fall study.

Comments on the Quality of English Language

the quality of English is low - a great rewriting is necessary followed by a English scientific editing.

Reviewer 3 Report

Comments and Suggestions for Authors

Dear Editor,

The present research manuscript titled ‘Comparative analysis of Microbial Communities in the Rhizosphere Soil and Plant roots of Healthy and Diseased Yuanyang Nanqi (Panax vietnamensis)’ expresses the challenges and problems in agricultural lands, especially at Yuanyang Nanqi. The current research manuscript emphasises the role of microbial communities in improving soil and plant root via its biological functions. It is a crucial topic, as microbial communities are an important factors that is critically involved in several soil-plant functions and acts as a limiting factor in plant growth and soil health improvement. The soil microbiome through direct antagonism and competition, as well as possible, suppresses plant diseases. Hence, there is a need to devise suitable ways for increasing its plant growth and improving soil health. The current research manuscript is very well written; however, following the nature of the comments, However, addressing several errors before final publication would improve the journal and manuscript quality, despite the interesting and novel findings. I will recommend minor revisions before acceptance.

General comments

There are some grammatical issues, and the English needs polishing throughout the manuscript as per the requirements. The context of the research is relevant to the Agricultural Journal and the findings will benefit for agricultural ecosystems and enhance the literature.

Specific comments

1.     In the abstract, conclusion portion: Try to extract the most interesting outcome of the study and to connect your research to the greater questions.

2.     Replace the keyword Microbial communities instead of bacterial communities, as paper is indicating both fungi and bacteria.

3.     Write something specific and more about comparative analysis.

4.     Please mention why only Panax vietnamensiswas used, even though other options are also available?

5.     If still authors has a sample, please check the fungal CFUs from practices soils and make a better comparison based on microbial biomass. Which is very important for the soil health indication.

6.     Results are almost fine, no need to change as per my knowledge.

7.     Overall, I am satisfied with the startup of discussion, but needs to add more latest references and make better comparison.  

8.  Body reference and main reference styles are not as per journal requirements; even the body reference wording style is also different. The author must check the references before submitting the manuscript.

Best wishes

Comments on the Quality of English Language

Need to be improve 

Round 2

Reviewer 1 Report

Comments and Suggestions for Authors

I have read the revised paper and believe that significant improvements have been made by the authors..  I believe the paper is now  ready to be published.

Comments on the Quality of English Language

The English is understandable.